**Data Availability Statement:** All relevant data are within the manuscript and its Supporting Information files.

# Multi-level determinants of land use land cover change in Tigray, Ethiopia: A mixed-effects approach using socioeconomic panel and satellite data

**Tadele Tafese Habtie**[1]*, **Ermias Teferi**[1], **Fantu Guta**[2]

1 Centre for Environment and Development, College of Development Studies, Addis Ababa University, Addis Ababa, Ethiopia, 2 Department of Economics, College of Business and Economics, Addis Ababa University, Addis Ababa, Ethiopia

* tadeleth74@gmail.com

## Abstract

This study examined land use land cover change and its determinants in Tigray, Ethiopia and its livelihood zones. We used socioeconomic panel, and satellite data, and applied a mixed-effects model to analyse the factors influencing land allocation among different uses, and transition matrix to analyse land cover dynamics. The results revealed that; land use choices were influenced by plot level factors (such as plot elevation, distance, soil type and quality, and land tenure), household characteristics (such as education, dependency ratio, plot size and number owned, income, livestock and asset, perception of climate change, and access to market and main road), and community level factors (precipitation, product price, population density and livelihood zone variations). Transition matrix analysis showed that between 1986 and 2016, 12.8% of forest was converted to bare land, 6.26% bare land was converted to pasture, and 5.84% of cropland was converted to forest. However, net deforestation occurred in most of the livelihood zones. Therefore, local communities faced environmental and socio-economic challenges from capital constraints induced land fallowing, land fragmentation, and unmanaged land cover change. The study recommended sustainable land use planning and management, market linkages, improved access to roads, forestry subsidies, land tenure security, and land consolidation programs.

## Introduction

Land is a crucial resource that supports human existence and development [1]. However, human activities have caused changes to the land system that impact climate, ecosystem services, and land degradation [2–5]. Unmanaged land cover transitions increase the vulnerability of human-environment systems and threaten human well-being [6,7]. Sustainable development promotes environment-neutral economic interests to manage forests, halt and reverse land degradation and biodiversity losses, and promote sustainable use of terrestrial ecosystems [8].

**Funding:** The author(s) received no specific funding for this work.

**Competing interests:** The authors have declared that no competing interests exist.

However, in Ethiopia, continuous land use changes have resulted in deforestation and land degradation mainly due to agricultural expansion, overgrazing, deforestation, and settlement development [9,10]. This has led to changes in land cover, which are aggravated by the ever-growing population and increased demands for ecosystem services [11]. One of the most affected regions in Ethiopia is Tigray in the north, which is highly degraded and suffers from low agricultural productivity, food insecurity and poverty [12–14].

Most previous studies on Land Use Land Cover (LULC) change in Ethiopia have focused on change detection and/or macro-level determinant analysis [15,16]. However, few studies have considered both macro-level and household-level drivers of LULC change [17,18]. More-over, most studies have used cross-sectional data and aggregated analysis units, which may not capture the spatial and temporal heterogeneity of LULC dynamics [19,20].

Similarly, most LULC studies in Tigray have only examined LULC change detections, with-out identifying the proximate causes of land use change at different levels [9,21–23]. These studies combined Landsat satellite imagery and qualitative data from interviews, FGDs, or document review, and used methods such as visual comparison, matrix analysis, and descriptive analysis to indicate the drivers of change. However, none of these studies, including [24,25], considered the household-level and plot level factors, used panel data to track the temporal dynamics of land use choices, or disaggregated their analysis by livelihood zone (LHZ), even though they tried to use logistic regression model to identify the macro level determinants of land use. Therefore, a more comprehensive and detailed analysis of LULC change and its drivers in Tigray is needed.

To address this research gap, this study aims to examine determinants of land use choices and land cover dynamics in Tigray using socioeconomic panel and satellite images. The novel contributions of this study include: (1) we use a hierarchical model that examines how factors at different levels (plot, household, and community and above) affect land use choice, (2) we analyze land use changes at the plot level within households, (3) we use panel data with repeated measurements for the year 2012, 2014, and 2016 to track how land use choices change over time, and (4) we disaggregate our analysis by LHZ, which is a smaller and more homogeneous unit with regard to livelihoods than river basin or sub-basin. This paper is organized as follows: Section 2 describes the materials and methods used in this study. Section 3 depicts the results. Section 4 presents the discussion and Section 5 concludes.

## Background

Tigray is a northern Ethiopian region with a population of about 6.1 million. It covers an area of roughly 53,000 km$^2$ and has diverse landscapes, climates, and livelihoods. It has three main agro-climatic zones (lowland, midland, and highland) and 16 LHZs, with elevations ranging from 500 to 3200 meters above sea level (Fig 1). The average annual temperature in Tigray varies from 13–28°C, while the average annual rainfall is between 388–602 mm [26].

The region has a diverse and complex agro-ecological system that supports various livelihood activities. According to the Famine Early Warning Systems Network (FEWS NET), the region is divided into 16 LHZs, which are geographical areas where people have similar patterns of access to food, income, and markets [27]. The main sources of food and income in each zone are rain-fed or irrigated agriculture and livestock rearing. The region produces cereals (such as teff, wheat, barley, sorghum, and maize), pulses, oilseeds (such as sesame and nug), and gesho (a plant used for making local beer) as the main crops. The main livestock in the region are cattle and shoats (sheep and goats). S1 Table provides more details on the features and descriptions of each livelihood zone.

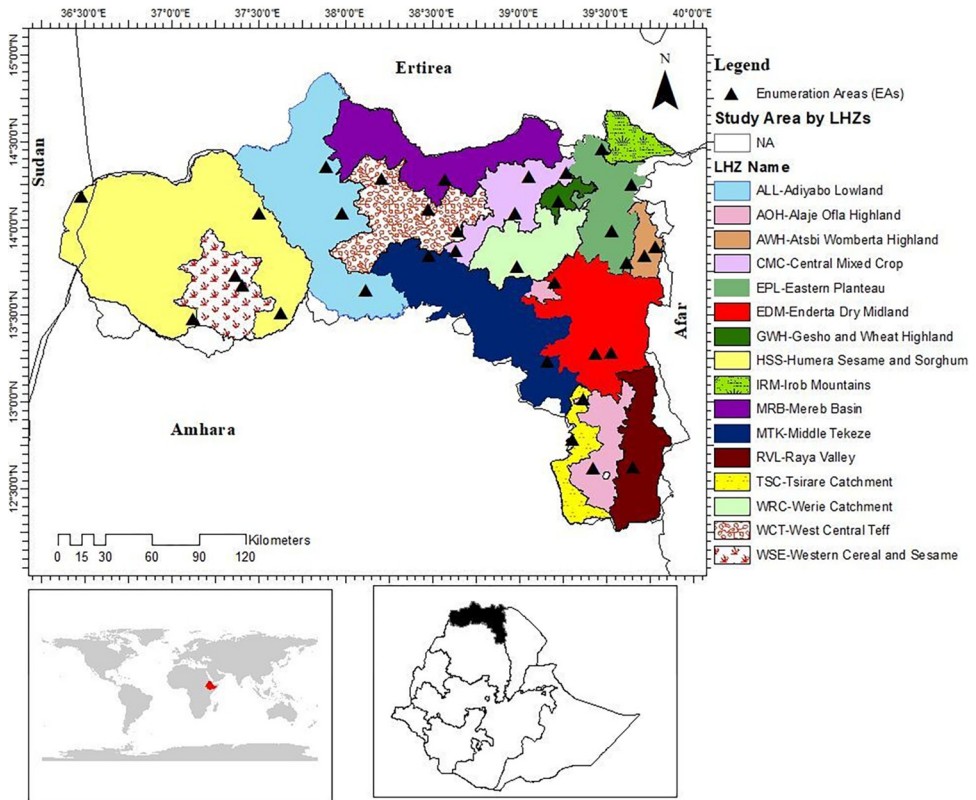

**Fig 1. Map of Tigray in Ethiopia.** The map shows the location, livelihood zones (LHZs), and enumeration areas (EAs) of Tigray. The LHZ is a spatial classification based on the dominant livelihood activities and agro-ecological conditions [27]. The EA is the primary sampling (smallest geographic) unit, used by the Central Statistics Agency (CSA) of Ethiopia for survey sampling and data availability [28–30]. Source: Author's own work using ArcGIS software and spatial data on livelihood zones from Famine Early Warning Systems Network (FEWS NET) [https://fews.net] [27] and on Enumeration Area (EA) from CSA of Ethiopia [28].

Agriculture is, then, the mainstay of Tigray's economy, accounting for about 36.7% of the regional GDP and employing about 83% of the workforce. It is based on a mixed farming system that combines crop and livestock production on the same farm, with 81.4% of farmers engaged in crop production and 14.5% in livestock production [31]. However, the region is challenged by environmental degradation, climate change, food insecurity, and poverty. Deforestation and land degradation have been affecting the region for decades, causing severe soil erosion and yield reduction [12,32,33].

As a result, Tigray is regarded as one of the most drought-prone and food-insecure areas in Ethiopia with frequent droughts occurring every two to four years [34–36]. To address these challenges the government has implemented various policies and programs such as the Productive Safety Net Program (PSNP), Soil and Water Conservation initiatives (a 20-day/year campaign of all citizens regardless of their PSNP status) and community-based natural resource management approaches (including exclosures and cropland afforestation). These programs have achieved significant success stories that have been recognized internationally [37–39]. However, despite these efforts, there is still a lack of understanding of the underlying causes and dynamics of LULC change in Tigray at different levels. Previous studies have mainly focused on detecting LULC changes using satellite imagery and qualitative data, but have not considered the household-level and plot-level factors, the temporal variations of land use choices, or the differences across livelihood zones. Therefore, this study aims to fill this gap

by conducting a more comprehensive and detailed analysis of LULC change and its drivers in Tigray.

## Materials and methods

### Data and data source

**Socioeconomic data.** This research used data from the Ethiopia Rural Socioeconomic Survey (ERSS), which is part of the World Bank's Living Standards Measurement Study, Integrated Survey on Agriculture (LSMS-ISA) project. The ERSS is integrated with the Central Statistical Agency's (CSA) Annual Agricultural Sample Survey (AgSS), in which the ERSS sample households are a subsample of the AgSS sample households, selected using a two-stage stratified random sampling method. The ERSS/LSMA-ISA sample is representative at the national and regional levels.

The ERSS/LSMA-ISA aims to improve the understanding of the links between agriculture and poverty reduction in Ethiopia by providing high-quality and timely data on the socioeconomic conditions of households, especially in relation to their agricultural activities and outcomes. The ERSS/LSMA-ISA was conducted in three rounds over 2012, 2014, and 2016, with two visits per year, one in the post-harvest season and one in the pre-harvest season. The ERSS/LSMA-ISA collects detailed information on various aspects of rural livelihoods, such as land use, land characteristics, and agricultural practices, household demographics, education, health, labor, consumption, income, assets, agriculture, shocks, food security, and social networks, as well as links plot and community level geo-referenced geospatial data (measures of distance, climatology, soil and terrain, and other environmental factors). The ERSS/LSMA-ISA data, along with its documentation can be accessed for free from the [Data Catalog (worldbank.org)]. For more details on the ERSS/LSMA-ISA overview, questionnaires, data dictionary please refer [LSMS-ISA (worldbank.org)].

This analysis relied on data from 996 households and 2,211 plots (with total of 7,492 fields) in Tigray, a region in northern Ethiopia. The analysis sample consisted of 387 households that were surveyed repeatedly in 37 EAs in all three rounds in rural areas and small towns, where the population is expected to predominantly live on agriculture. Additionally, the analysis conducted a follow-up survey on public climate and LULC change perceptions and opinions with the 387 households who participated in the third round of ERSS/LSMS-ISA data collection, to complement the ERSS/LSMS-ISA data [refer S4 Appendix]. This follow-up survey was conducted in August, 2019.

**Satellite image processing and classification.** On-demand Landsat images for 1986 and 2016 from Landsat 5 Thematic Mapper, and Landsat 8 Operational Land Imager, respectively, were used. We, then, categorized the region's LULC categories into Cropland, Bare land, Forestland, Pasture land and Other LULC class to fit the existing LULC class in ERSS/LSMS-ISA dataset. The operational definition of each LULC category is given in (Table 1). The choice of years considers major event experienced in the region (famous drought in 1986), the last year of ERSS/LSMA-ISA panel date and data access challenges (difficulties in validating satellite imagery data for additional years due to the large area coverage and security challenges). All Landsat images were accessed free of charge from the US Geological Survey Center for Earth Resources Observation and Science via Earth Explorer (https://earthexplorer.usgs.gov/). Essential pre-processing operations such as radiometric and atmospheric corrections (Haze reduction) were performed using Arc GIS 10.5 and ERDAS Imagine 14 software. The radiometric correction of Landsat satellite sensors' Digital Number (DN) values were transformed into radiance and subsequently into top-of-atmosphere (TOA) reflectance [40].

**Table 1. Summary of land use land cover categories and descriptions.**

| Category | Description |
|---|---|
| Bare land | Degraded or exposed land devoid of vegetation and mainly includes (mining and construction sites), sands exposed to the surface, non-agricultural bare soils and rocks found in steep slope areas and along river banks |
| Forest land | Natural and plantation forests found in the study area include woodlands, shrubs and bushlands that are under government and communal protection and agroforestry areas. |
| Pasture land | Landscapes in which grasses are the dominant vegetation types or areas dominated by natural grass, including areas used for traditional grazing |
| Crop land | Arable and fallow land grows annual or perennial crops through rain-fed and small-scale or commercial irrigation. |
| Other land uses land cover | Areas covered by water, wetlands, and settlement |

We also used ancillary data to improve classification accuracy and support interpretation of land cover changes. LHZ data were obtained from Famine Early Warning Systems Network (https://fews.net/data/livelihood-zones). Digital Elevation Model (SRTM-30m) and its derived datasets such as slope were used to supplement satellite data in mapping land cover, as vegetation classes are often limited to specific agro-climatic conditions. DEM of SRTMGL1 (NASA Shuttle Radar Topography Mission Global 1 arc second (~30 m) V003) was obtained from http://e4ftl01.cr.usgs.gov/SRTM/SRTMGL1.003/.

A hybrid supervised/unsupervised classification approach was implemented together with successive Geographic Information System operations (spatial analysis) to classify the Landsat imageries based on homogenous land units. First, to determine the spectral classes Iterative Self-Organizing Data Analysis clustering was performed [41]. Second, ground truth data was collected from field observations as well as Google Earth to associate the spectral classes with the cover types of the already defined classification scheme. Ground truth data for LULC classification and accuracy assessment were collected using different information. The reference data for the 1986 image was gathered by interviewing local elders and using some old toposheets from 1993 and Google Earth to identify known locations in the area. It's been found that information collected from elderly people using a participatory approach is a reliable source of reference data for validating old land use and land cover maps [42]. To determine the historical land cover status of each sample point, 34 key informant interviews conducted in 2019 and old toposheets were used. For the 2016 image, Google Earth's time slider image was used as reference truth points to gather the necessary reference data. A total of 375 training points were collected using a simple random sampling technique from the five major representatives of LULC classes for each study year. Finally, images were classified using Maximum Likelihood Algorithm based on the training samples developed in the previous step [43]. In order to use the classified maps for further LULC analysis, the classification accuracy was assessed. To ensure accuracy, the thumb rule of Congalton [44] was applied to each LULC class, with at least 75 samples (pixels) collected. A stratified method of random sampling was used to collect sample reference points from individual land cover strata. The number of samples was selected for each map class by assigning equal sample sizes to all strata to favor estimating the accuracy of the user overestimating the overall and producer accuracies. Overall accuracies of 87% and 89% were attained for the 1986 and 2016 classified maps, respectively (S2 Table).

## Method of data analysis

We want to examine determinants of land use choices and land cover dynamics in Tigray, a region where agriculture is the main human activity. Land use allocation is a complex and

multidisciplinary process that depends on various biophysical and socioeconomic factors that vary across time, space, and decision-making scales [5,45]. To capture this complexity and diversity, a plot-level approach is required that can account for the heterogeneity and dynamics of these factors [46]. However, such an approach faces data availability, scale mismatch, and disciplinary perspective challenges [46,47]. This study overcomes these challenges by applying a mixed-effects model of land use allocation at the plot level, which is the smallest spatial unit of land management by a farming household. Furthermore, this study uses a multi-method land cover change detection procedure to examine the main trajectories of land cover change. S1 Fig depicts the methodology we followed. The following sub-sections presented theoretical and empirical models for the land use choice analysis and methods of land cover change detection adopted.

**Theoretical model.** Literature on the spatial land allocation theory [48–50], postulates that land is divided between alternative uses according to the highest land rent obtainable, given the land's special characteristics with regards to location and physical conditions. However, in Ethiopia, land is publicly owned and hence land markets do not exist effectively and land prices reflecting the land rent of a particular land quality class in alternative uses are not available [51].

The existence of capital constraints and imperfect markets in Ethiopia [52,53] further complicates land use and production decisions. Despite these challenges, it can be postulated that farming households in Tigray, who have been granted land use rights, will allocate their land to different uses based on the profitability of each use category [54–58]. This allocation is influenced by the production and sale of their outputs in the existing farm output and input markets, the market prices and demand, and the receipt of input supports and government intervention to mitigate capital constraints and imperfect markets [59–62].

Hence, analysis of the behaviour of land use allocation in Tigray for this study is based on a profit-maximization theoretical model derived in the S1 Appendix. In this modelling it is assumed that farmers will allocate their homogenous land area among different uses according to the profitability of each use, given their constraints and preferences.

**Empirical model and estimation.** This study specifies a linear equation for each land use/cover choice: which depends on the profitability of each use, given the constraints and preferences of farmers. The profitability of each use is determined by the community level price of each land use output, the characteristics of each plot allocated to each land use, and the household and community level factors that affect land use and production decisions [46].

The dataset used for this study is a plot-level three-round panel dataset, which contains information on the area allocated to each land use (arranged in a nx1 vector grouped by land use type), and on the prices, plot characteristics, and household and community factors (arranged in a nxm matrix of independent variables with block diagonal structure) for a sample of farmers in Tigray region. Land uses in this study are main land purposes of plots (cropland, forest, pasture, fallow and other uses) as outlined in the ERSS/LSMS-ISA dataset. We classify the observed plots of land according to how many times (rounds) they have been assigned to a specific land use category during 2012 to 2016. A plot can have one, two, or three observations, depending on how often it has changed its land use category. The number of observations indicates the degree of sustainability of the plot in a given land use category, which we call the sustainability group. The dataset has then made to have a hierarchical structure: plots are nested within sustainability groups, and sustainability groups are nested within land use types. Such structure of the dataset allows the study to control for unobserved heterogeneity across plots and sustainability groups, and to estimate separate land use/cover equations.

The study assumes that the error terms are correlated across different land uses, sustainability groups, and plots due to the competition for land among different uses and the unobserved

plot specific heterogeneity [56,63,64]. The study estimates a three-level mixed-effects model using the maximum likelihood method [65,66]. The model accounts for the correlation of error terms within plots over sustainability group and across different land uses. The model also captures the unobserved heterogeneity at different levels of the hierarchy, sustainability group and plot. Our procedure estimated each land use equations (cropland, pasture, fallow, forest, and other land uses) separately and software used for estimation is Stata version 17. For more details on the empirical model specification and estimation, please refer to S2 Appendix.

**Land use/cover change detection.** Though the land use allocation specified as mixed-effects model can capture high level of explanation of recent changes in land use, it may fail to capture the long-term trends and patters of land use change due to its low temporal complexity covering only a short duration (5 years) and a few time steps (3 years) [50]. Therefore, we complemented the model with a historical analysis of land cover spatial data, derived from satellite imagery for the periods 1986 and 2016. We used a similar land cover classification scheme that matches the land use classification of plots in the ERSS/LSMA-ISA dataset. However, the land use types (plot level) are also in a very low spatially resolution, and they may not correspond to land cover types that are identified through satellite image processing and remote sensing. We assumed a dominant land cover change process in cases of resolution mismatch between the plot and pixel data [67].

We used two methods in broad to detect and analyse LULC change in Tigray between 1986 and 2016: Growth rate and transition matrix analysis. They are suitable for LULC change analysis, as they can handle complex and non-linear phenomena, such as land cover dynamics [20,68]. Growth rate measures the rate of change of each LULC category over time, and transition matrix analysis shows the proportions of each LULC category that changed or persisted over time. Using the transition matrix analysis, we have conducted: component of change analysis, vulnerability to change analysis, and key signal of change analysis. Component of change measures the gross loss, gross gain, net change, and swap change for each category. Vulnerability to change measures the persistence indices for each category to show their vulnerability to loss, gain, or net change. Key signal of change identifies systematic transitions that deviate from random processes. These techniques are based on the approaches of Briones & Varas [69], Ewunetu et al. [20], Pontius et al. [70], and Alo & Pontius Jr [71]. For details of the arguments and mathematical formulations of each method and technique, please refer to S3 Appendix.

## Results

### Land use choice analysis

We analyse land allocation to land uses. We describe our model variables. Land without market value is allocated based on profitability which depends product prices [55,72]. Land allocation also depends on land conditions, and socioeconomic and environmental factors [54,57].

**Descriptive statistics.** Table 2 depicts description of the main variables of the study. About 72% Tigray farmers who decide land allocation are illiterate and have one economically active for each dependent in their five-member households. However, 77% of farmers also engage in non-farm activities. Their main livelihood is farming, owning and cultivating 87% of their land on an average of two to three plots in different locations. In addition, they live on average 1 km away from their plots, which 46% have fair soil quality, 21% are vertisols type, and have high elevation and moderate slope. Despite their low asset ownership and income, they benefit from extension services and have experienced income growth in 2012 to 2016. Nevertheless, they face challenges such as long distances to markets and roads, unstable prices,

**Table 2. Descriptive statistics for discrete and continuous descriptors by panel rounds and pooled.**

| Variables | Pooled | | Round | | | | | |
|---|---|---|---|---|---|---|---|---|
| | | | I | | II | | III | |
| | Mean | SD | Mean | SD | Mean | SD | Mean | SD |
| **Discrete** | | | | | | | | |
| *Plot descriptors* | | | | | | | | |
| Soil type is vertisols (Yes = 1) | 21.53 | - | 22.18 | - | 21.32 | - | 21.11 | - |
| Soil quality: Good | 26.16 | - | 27.94 | - | 26.30 | - | 24.39 | - |
| Soil quality: Fair | 46.34 | - | 42.53 | - | 42.56 | - | 53.26 | - |
| Soil quality: Poor | 27.50 | - | 29.53 | - | 31.14 | - | 22.36 | - |
| Land tenure: owned | 95.18 | - | 94.48 | - | 95.66 | - | 95.41 | - |
| Land tenure: Tenanted | 1.87 | - | 2.07 | - | 2.06 | - | 1.51 | - |
| Land tenure: Sharecropped | 2.19 | - | 1.14 | - | 2.28 | - | 3.09 | - |
| Land tenure: Irregular cultivated | 0.76 | - | 2.32 | - | 0.00 | - | 0.00 | - |
| *Household descriptors* | | | | | | | | |
| Literate household head (Yes = 1) | 27.91 | - | 26.06 | - | 25.07 | - | 32.20 | - |
| Access to extension (Yes = 1) | 98.09 | - | 97.39 | - | 98.21 | - | 98.59 | - |
| Nonfarm participation (Yes = 1) | 77.01 | - | 82.41 | - | 70.45 | - | 78.53 | - |
| **Continuous** | | | | | | | | |
| *Plot descriptors* | | | | | | | | |
| Average plot slope (%) | 12.62 | 10.96 | 13.31 | 11.37 | 12.57 | 11.02 | 12.03 | 10.48 |
| Average plot elevation (m) | 1943.08 | 540.17 | 1960.48 | 538.09 | 1922.50 | 536.61 | 1945.33 | 544.85 |
| Plot Dist. To residence (kms) | 1.02 | 15.55 | 1.24 | 26.79 | 0.96 | 3.67 | 0.87 | 2.28 |
| *Household descriptors* | | | | | | | | |
| Household size | 4.92 | 2.28 | 4.95 | 2.23 | 4.95 | 2.29 | 4.88 | 2.31 |
| Dependency ratio | 1.16 | 0.90 | 1.20 | 0.95 | 1.13 | 0.88 | 1.14 | 0.88 |
| Dist. to the nearest market (kms) | 57.87 | 27.42 | 58.43 | 27.37 | 57.37 | 27.38 | 57.87 | 27.56 |
| Dist. to nearest Major Road (kms) | 17.98 | 16.50 | 17.97 | 16.56 | 18.14 | 16.45 | 17.85 | 16.53 |
| Asset index | 0.08 | 0.13 | 0.02 | 0.07 | 0.09 | 0.12 | 0.12 | 0.15 |
| Annual income (00 Birr) | 241.95 | 950.08 | 64.08 | 164.85 | 149.45 | 539.48 | 483.73 | 1466.66 |
| Livestock (TLU) | 3.58 | 4.52 | 3.28 | 3.45 | 3.03 | 3.38 | 4.36 | 5.97 |
| Total size (hectare) | 0.51 | 0.19 | 0.50 | 0.19 | 0.52 | 0.21 | 0.52 | 0.23 |
| Proportion of crop land | 0.87 | 0.18 | 0.88 | 0.18 | 0.87 | 0.17 | 0.86 | 0.20 |
| Proportion of pasture land | 0.10 | 0.10 | 0.10 | 0.10 | 0.11 | 0.12 | 0.08 | 0.08 |
| Proportion of fallow land | 0.24 | 0.23 | 0.20 | 0.21 | 0.27 | 0.22 | 0.26 | 0.25 |
| Proportion of forest land | 0.07 | 0.09 | 0.05 | 0.05 | 0.08 | 0.07 | 0.09 | 0.13 |
| Proportion of other land uses | 0.15 | 0.28 | 0.12 | 0.23 | 0.16 | 0.30 | 0.17 | 0.31 |
| Average no of plots owned | 2.52 | 0.53 | 2.24 | 0.74 | 2.18 | 0.74 | 2.2 | 0.83 |
| Climate change risk perception | 0.24 | 0.36 | 0.14 | 0.30 | 0.17 | 0.33 | 0.40 | 0.39 |
| *Community descriptors* | | | | | | | | |
| Average cropland output price* | 65.29 | 184.66 | 90.57 | 317.29 | 10.26 | 9.38 | 94.52 | 30.67 |
| Average pasture land output price* | 14.23 | 16.95 | 20.03 | 24.61 | 23.61 | 9.70 | 2.56 | 1.02 |
| Average forest land output price* | 229.88 | 197.05 | 73.75 | 26.16 | 100.07 | 26.01 | 442.37 | 135.80 |
| Population density (per km$^2$) | 738.7 | 2586.66 | 579.15 | 2026.4 | 747.91 | 2578.5 | 868.36 | 2994.99 |
| Proportion of Agr. within 1 km buffer | 0.36 | 0.23 | 0.37 | 0.23 | 0.35 | 0.22 | 0.36 | 0.22 |
| Annual mean precipitation (mm) | 701.74 | 104.78 | 702.75 | 105.15 | 701.85 | 106.69 | 700.75 | 102.91 |
| **Households** | **996** | | **307** | | **335** | | **354** | |

(*Continued*)

**Table 2.** (Continued)

| Variables | Pooled | | Round | | | | | |
|---|---|---|---|---|---|---|---|---|
| | | | I | | II | | III | |
| | Mean | SD | Mean | SD | Mean | SD | Mean | SD |
| **Plots** | **2,211** | | **689** | | **730** | | **792** | |

Note: This table shows the percentage for discrete descriptors and the mean and standard deviation for continuous descriptors. Climate change perception index is constructed by combining individuals' climate change beliefs, personal experience with extreme weather events, its frequency, severity, and associated livelihood losses. Dependency ratio is derived by dividing the number of dependents (people under 18 or over 64) by the number of working-age people (people between 18 and 64) in the household. Soil quality reflects the household's assessment of the soil quality as good, fair, or poor. Soil type follows the FAO classification of soil as leptosol, cambisol, vertisol, luvisol, mixed type, and other soil type.

*Prices are quantity weighted average producer prices (Birr/kg). For crop land, products were mainly cereals; for pasture, meat, milk & milk products; for forest, fuelwood, charcoal & honey.

climate change and high population density. Moreover, the region falls below the national average of annual rainfall.

**Determinants of land use allocation.** We used a mixed model to estimate five land use equations with Stata. Table 4 shows the maximum likelihood results. The LR and Wald tests support a three-level mixed model (intercept heterogeneity) and a joint fit [73]. The AIC in Tables 4 and S3 showed the robustness of results of mixed model compared to separate OLS for each land use equations. Land use equation residuals are also found correlated supporting our specification (Table 3).

The results (Table 4) show that plot-level factors such as elevation, distance to residence, soil type and quality, and tenure affect land use in Tigray. Higher altitude increases fallow and pasture. Longer distance reduces forest and other land uses. Vertisols soil increases fallow and decreases forest land use compared to other types. Fair or poor soil quality reduces cropland and other land uses compared good soil quality while pasture is negatively related to poor soil quality. Tenanted land increases cropland and other land uses compared to owned land.

Household-level factors also affect land allocation. Increase in plot size increases all land uses, while increase in number of plots decreases them. Increase in dependency ratio increases fallow and decreases cropland and other land uses. Being literate head and increase in asset index and climate change perception decrease cropland, while higher asset index increases other land uses. The longer residence distance to market decreases cropland, fallow, and pasture, while longer distance to road decreases other land uses. Livestock ownership and income decrease pasture and forest, respectively.

Community-level factors such as forest product price, precipitation, population density, and livelihood zone variations influence land use. Forest product price increases forest, while precipitation decreases it. Population density increases other land uses. Eastern Plateau (EPL), compared to other livelihood (OTH), allocate less land to cropland while livelihood zone variations have mixed effect on allocation to others land use.

**Table 3. Correlation matrix of residuals of land use equations: After mixed model estimation.**

| | Cropland | Pasture | Fallow | Forest | Others |
|---|---|---|---|---|---|
| Cropland | 1.0000 | | | | |
| Pasture | 0.2207 | 1.0000 | | | |
| Fallow | -0.0939 | 0.0523 | 1.0000 | | |
| Forest | -0.1236 | -0.0774 | -0.3623 | 1.0000 | |
| Others | -0.0232 | -0.1778 | -0.1979 | -0.4334 | 1.0000 |

## Land cover change

**Spaciotemporal distribution of LULC categories.** The study showed significant changes in all landscapes over 30 years, with low persistence from 37.1% in Adiyabo Lowland (ALL) to 61.3% in Enderta Dry Midland (EDM) LHZs (Fig 2F). We calculated the long-term trend change of land cover in Tigray and its main LHZs and verified that all LULC types changed over time, except for cropland in Tigray, which seemed almost unchanged (less than 1% declined) in 2016 (Fig 2E). Tigray had less forest (18.7%) and bare land (12.8%) (Fig 2C and 2D), but more pasture (103.6%) and other land covers (84%) (Fig 2A and 2B). Forest degraded most in Humera Sesame and Sorghum (HSS) (42.8%) and ALL (29.2%) but recovered in

**Table 4. Mixed-effects model for land use share equations.**

| Land use equation | | Cropland | Pasture | Fallow | Forest | Others |
|---|---|---|---|---|---|---|
| *Plot level factors* | Slope (%) | 0.0010 | -0.0004 | 0.0007 | -0.0005 | -0.0005 |
| | | (0.0007) | (0.0007) | (0.0011) | (0.0007) | (0.0008) |
| | Elevation (m) | 0.00001 | 0.0001 | 0.0001 | 0.00004 | 0.00002 |
| | | (0.00002) | (0.00002) ** | (0.00003) ** | (0.00003) | (0.00002) |
| | Dist. To residence (kms) | 0.0003 | -0.0068 | -0.0006 | -0.0149 | -0.0072 |
| | | (0.0007) | (0.0076) | (0.0050) | (0.0064) ** | (0.0030) ** |
| | Soil type is vertisols (Yes = 1) | -0.0149 | 0.0026 | 0.0919 | -0.0316 | -0.0195 |
| | | (0.0136) | (0.0189) | (0.0297) *** | (0.0188) * | (0.0190) |
| | Soil quality: Fair | -0.0218 | 0.0032 | -0.0119 | | -0.0762 |
| | | (0.0131) * | (0.0131) | (0.0284) | | (0.0182) *** |
| | Soil quality: Poor | -0.0300 | -0.0257 | 0.0279 | | -0.0951 |
| | | (0.0149) * | (0.0139) * | (0.0318) | | (0.0203) *** |
| | Land tenure: Tenanted | 0.1104 | -0.0150 | -0.0661 | | 0.2996 |
| | | (0.0307) *** | (0.0504) | (0.0735) | | (0.0344) *** |
| | Land tenure: Sharecropped | 0.0204 | 0.0236 | -0.0594 | | 0.0531 |
| | | (0.0284) | (0.0382) | (0.0633) | | (0.0391) |
| *Household level factors* | Literate house head (Yes = 1) | -0.0245 | -0.0120 | | | 0.0194 |
| | | (0.0117) ** | (0.0124) | | | (0.0164) |
| | Dependency ratio | -0.0156 | 0.0038 | 0.0287 | | -0.0144 |
| | | (0.0057) *** | (0.0054) | (0.0112) ** | | (0.0079) * |
| | Ln of total area (hectare) | 0.0657 | 0.0660 | 0.1101 | 0.0413 | 0.0378 |
| | | (0.0045) *** | (0.0043) *** | (0.0096) *** | (0.0065) *** | (0.0085) *** |
| | No of plots owned | -0.0058 | -0.0058 | -0.0166 | -0.0056 | -0.0293 |
| | | (0.0014) *** | (0.0012) *** | (0.0022) *** | (0.0017) *** | (0.0016) *** |
| | Livestock (TLU) | | -0.0039 | | | |
| | | | (0.0015) *** | | | |
| | Asset index | -0.0940 | | -0.0861 | | 0.1412 |
| | | (0.0475)** | | (0.1219) | | (0.0624) ** |
| | Ln of annual income | 0.0031 | | | -0.0195 | |
| | | (0.0034) | | | (0.0053) *** | |
| | Dist. To nearest market (km) | -0.0005 | -0.0006 | -0.0026 | 0.0013 | 0.0005 |
| | | (0.0003) * | (0.0003) * | (0.0007) *** | (0.0021) | (0.0004) |
| | Dist. To Major Road (km) | 0.0006 | 0.0009 | 0.0013 | 0.0022 | -0.0018 |
| | | (0.0004) | (0.0006) | (0.0010) | (0.0029) | (0.0006) *** |
| | Climate change perception index | -0.0299 | | | -0.0135 | 0.0131 |
| | | (0.0154) * | | | (0.0253) | (0.0213) |

(*Continued*)

**Table 4.** (Continued)

| Land use equation | | Cropland | Pasture | Fallow | Forest | Others |
|---|---|---|---|---|---|---|
| ***Community level factors*** | Average output price (Birr) | -0.00003 | -0.0004 | | 0.0003 | |
| | | (0.00003) | (0.0003) | | (0.0001) *** | |
| | Population density (per km²) | -0.000002 | -0.00001 | 0.0010 | 0.000003 | 0.00002 |
| | | (0.000002) | (0.000005) | (0.00001) | (0.000004) | (0.00003) *** |
| | Annual mean Rainfall (mm) | | 0.0000 | | -0.0007 | |
| | | | (0.0001) | | (0.0004) ** | |
| | Prop. Of agri. In 1 km buffer | | | | -0.0864 | |
| | | | | | (0.0808) | |
| | Livelihood Zone: ALL | -0.0195 | -0.0152 | 0.0109 | | -0.0885 |
| | | (0.0235) | (0.0297) | (0.0490) | | (0.0326) *** |
| | Livelihood Zone: CMC | 0.0334 | 0.0225 | -0.0645 | -0.0504 | -0.0386 |
| | | (0.0205) | (0.0251) | (0.0537) | (0.1280) | (0.0293) |
| | Livelihood Zone: EDM | 0.0245 | 0.0344 | -0.0959 | 0.0306 | -0.0779 |
| | | (0.0233) | (0.0514) | (0.0807) | (0.0929) | (0.0324) ** |
| | Livelihood Zone: EPL | -0.0547 | -0.0258 | -0.0753 | 0.0112 | 0.0955 |
| | | (0.0224) ** | (0.0193) | (0.0464) | (0.0881) | (0.0294) *** |
| | Livelihood Zone: HSS | -0.0434 | 0.0307 | 0.0525 | | -0.0872 |
| | | (0.0267) | (0.0348) | (0.0579) | | (0.0359) ** |
| | Livelihood Zone: WCT | -0.0075 | -0.0087 | 0.0686 | 0.0085 | 0.0766 |
| | | (0.0229) | (0.0220) | (0.0489) | (0.0757) | (0.0317) ** |
| | Constant | 0.9957 | 0.3053 | 0.5825 | 0.7497 | 0.5368 |
| | | (0.0545) *** | (0.0888) *** | (0.1114) *** | (0.2588) *** | (0.0720) *** |
| Observations = 2.211 | | ᶻWald Chi² = 34141.47 *** | | ˚LR = 444.36 *** | | AIC = -1612.53 |

Note: This table shows the coefficients and standard errors for variables at plot, household and community levels that affect the land use shares. ALL, Adiyabo Lowland; CMC, Central Mixed Crop; Enderta Dry Midland; EPL, Eastern Plateau; HSS, Humera Sesame and Sorghum; WCT, West Central Teff.

ᶻJoint model fitness Chi² test.

˚LR test vs. linear model (Chi² test).

* p<0.1

** p<0.05

*** p<0.01, Standard errors are in parenthesis.

Central Mixed Crop (CMC) (27.2%) (Fig 2C). Pasture expanded most in HSS (threefold) and ALL (twofold) (Fig 2A), while cropland expanded only in HSS (20.8%) and EDM (18.7%) (Fig 2E). Significant expansion (about 3 to 4 times) in other land covers was also observed in HSS, CMC, and West Central Teff (WCT) LHZs (Fig 2B).

**The LULC inter category transition.** *Component of change.* We quantified landscape transitions with respect to persistence, using gross losses and gains, net change, swap, and total change. These components were derived from the transition matrices (S4 Table) and shown in Fig 3. In Tigray, bare land and forest experienced the highest total change, with 34.1% and 30.7%, respectively, followed by cropland (14.1%). However, most of the land cover change in Tigray was swap dominated, which balanced out the losses and gains of different land cover types, resulting in lower net loss of 5.7% and 4.8% in forest and bare land (Fig 3C and 3D) and lower net gain of 5.6% and 5.2% in pasture and other land uses (Fig 3A and 3B). Moreover, dynamics in crop land in Tigray was mainly due to swamp resulting in insignificant net change (Fig 3E).

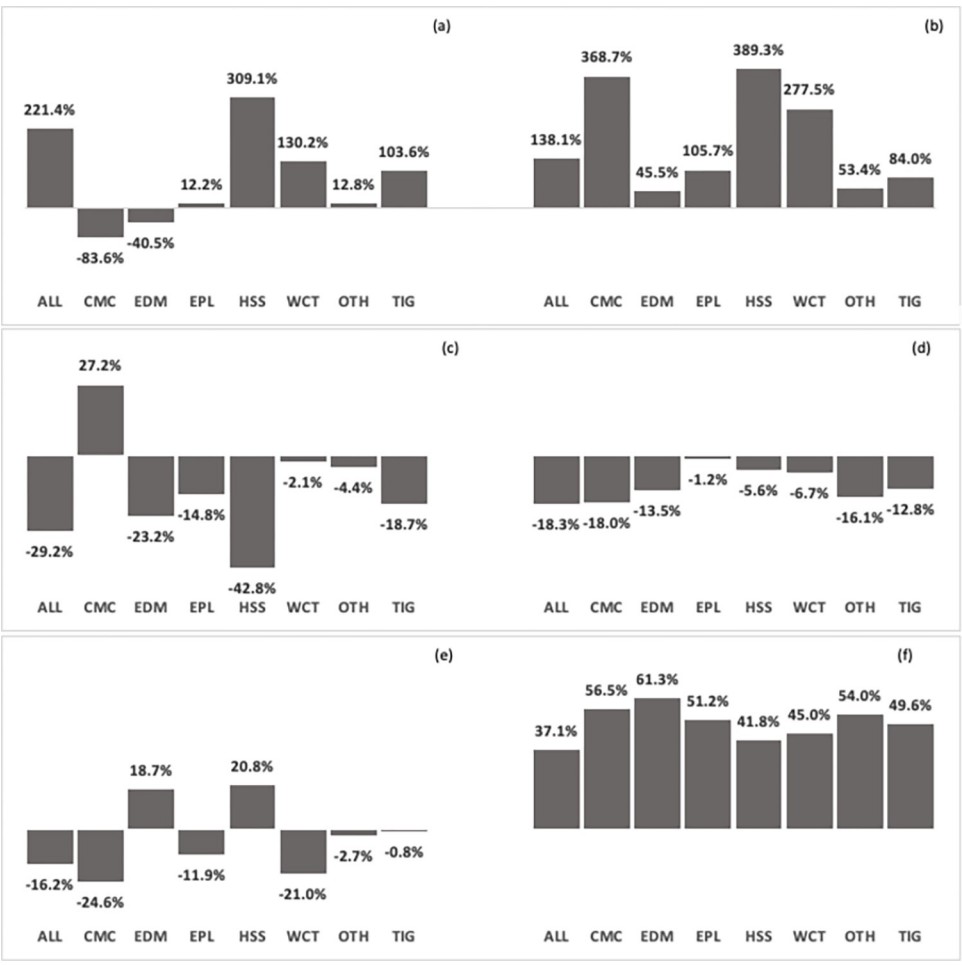

**Fig 2. Persistence and land cover trend change by LULC category in Tigray and its main LHZs during 1986 to 2016.** (a) Trend change for pasture (b) Trend change for other land covers (c) Trend change for forest (d) Trend change for bare land (e) Trend change for cropland (f) Persistence of land cover categories. ALL, Adiyabo Lowland; CMC, Central Mixed Crop; Enderta Dry Midland; EPL, Eastern Plateau; HSS, Humera Sesame and Sorghum; WCT, West Central Teff; OTH, other land uses; TIG, Tigray. Source: Author's own work using ArcGIS and LandSat imagery.

LULC dynamics across LHZs showed a mixed pattern. Forest was the most dynamic in WCT (40.3%) and HSS (38%) (Fig 3C), while bare land had highest total change in ALL (45.2%), and HSS (44.8%) (Fig 3D). Pasture was the most dynamic in ALL (25.4%) and HSS (21.1%) (Fig 3A), and higher total change was observed in other land uses in EPL (15.1%) and EDM (13.9%) (Fig 3B). In EPL and WCT, cropland had highest total change of 21.8% and 19.1%, respectively (Fig 3E). Most (at least 86%) farming households in each LHZ also perceived such dynamism in cropland, pasture, and forest.

*Vulnerability of LULC to dynamic.* Table 5 shows persistence indices of LULC dynamics in Tigray. Except for cropland (relatively stable in net change), all categories tended to transition from loss/gain to another category. Forest and bare land lost more area, while pasture and other land uses gained more area than they persisted. Pasture gained the most area (a total increment of 4.4 times or a net gain of 2.7 times its persistence), followed by other land uses (a total increment of twice or a net gain of 1.3 times its persistence). Forest lost the most area (150% lower than its persistence) compared with the rest of the categories, followed by bare land (110% lower than persistence).

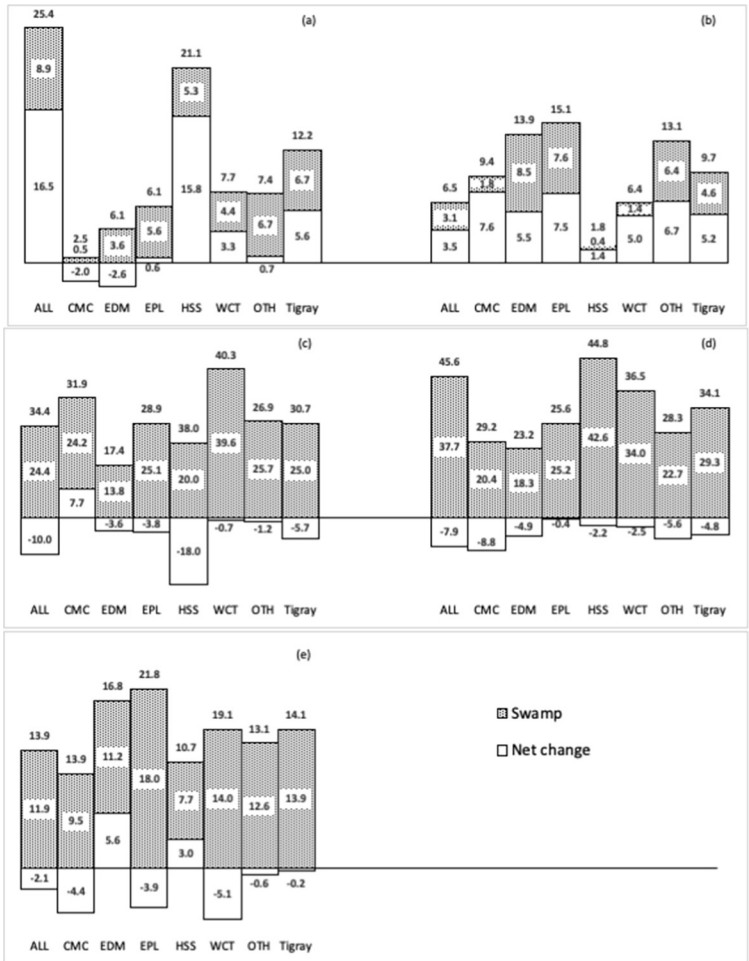

**Fig 3. Percentage net change, swamp change and total change of LULC category in Tigray and its main LHZs during 1986 to 2016.** (a) for pasture (b) for other land covers (c) for forest (d) for bare land (e) for crop land. Net percentage changes below the horizontal axis are net loss and above the vertical axis are net gain. ALL, Adiyabo Lowland; CMC, Central Mixed Crop; Enderta Dry Midland; EPL, Eastern Plateau; HSS, Humera Sesame and Sorghum; WCT, West Central Teff; OTH, other land uses; TIG, Tigray. Values at the top of each bar are total change which is the sum of swamp change and absolute value of net change. Source: Author's own work using ArcGIS and LandSat imagery.

LULC categories changed similarly across space (LHZs), except for forest and pasture in CMC. Forest gained more than it persisted (a total increment of 120% of its persistence), while pasture lost more than any other category (a total decrement of 17.7 times or a net loss of 15.6 times its persistence) in CMC (Table 5). Next to crop land, bare land had a lower tendency to change in all LHZs except in ALL, despite having the greatest loss/gain and the second highest total change. This inconsistency can be explained by the size of LULC categories and their persistence levels. Pontius et al. [70] suggested that the largest LULC categories may have large transitions by chance. We separated key and systematic patterns of change from any persistence level and land cover size to avoid misleading signals of transition and spurious landscape dynamics.

*Detecting key signals of LULC change.* Fig 4 shows how categories change based on gains and losses. Green up-arrows mean more positive changes than expected, red down-arrows mean more negative changes than expected, and yellow hyphens mean random changes. In

Table 5. Persistence indices for each LULC categories in Tigray and its main LHZs.

| Description | | Livelihood Zone (LHZ) | | | | | | | |
|---|---|---|---|---|---|---|---|---|---|
| | | ALL | CMC | EDM | EPL | HSS | WCT | OTH | Tigray |
| Pasture | Gain: persistent | 7.0 | 2.1 | 0.9 | 1.8 | 7.5 | 16.9 | 2.2 | 4.4 |
| | Loss: persistent | 1.5 | 17.7 | 2.2 | 1.5 | 1.1 | 6.8 | 1.8 | 1.6 |
| | Net change: persistent | 5.5 | -15.6 | -1.3 | 0.3 | 6.4 | 10.1 | 0.4 | 2.7 |
| Other uses | Gain: persistent | 5.1 | 7.5 | 1.2 | 3.4 | 11.7 | 5.0 | 1.1 | 1.9 |
| | Loss: persistent | 1.6 | 0.8 | 0.5 | 1.2 | 1.6 | 0.6 | 0.4 | 0.6 |
| | Net change: persistent | 3.6 | 6.7 | 0.7 | 2.3 | 10.1 | 4.4 | 0.7 | 1.3 |
| Forest | Gain: persistent | 1.0 | 1.2 | 1.4 | 1.4 | 0.7 | 1.5 | 1.0 | 1.0 |
| | Loss: persistent | 1.8 | 0.8 | 2.1 | 1.8 | 2.0 | 1.5 | 1.1 | 1.5 |
| | Net change: persistent | -0.8 | 0.5 | -0.7 | -0.4 | -1.3 | -0.1 | -0.1 | -0.5 |
| Bare land | Gain: persistent | 1.2 | 0.3 | 0.4 | 0.8 | 1.5 | 0.9 | 0.6 | 0.8 |
| | Loss: persistent | 1.6 | 0.6 | 0.6 | 0.8 | 1.6 | 1.1 | 1.0 | 1.1 |
| | Net change: persistent | -0.5 | -0.3 | -0.2 | 0.0 | -0.2 | -0.1 | -0.3 | -0.3 |
| Crop land | Gain: persistent | 1.3 | 0.5 | 0.5 | 0.5 | 0.7 | 0.6 | 0.5 | 0.5 |
| | Loss: persistent | 1.7 | 1.0 | 0.2 | 0.7 | 0.4 | 1.0 | 0.5 | 0.6 |
| | Net change: persistent | -0.4 | -0.5 | 0.2 | -0.2 | 0.3 | -0.4 | 0.0 | 0.0 |

ALL, Adiyabo Lowland; CMC, Central Mixed Crop; Enderta Dry Midland; EPL, Eastern Plateau; HSS, Humera Sesame and Sorghum; WCT, West Central Teff; OTH, other livelihood zones.

Tigray and most LHZs, only two changes were random: transition between "*pasture and forest*" and the loss of "*other land uses to bare land*,". In ALL, all changes were systematic, while in CMC, many changes were random, especially to "*pasture*". The most important systematic transitions (highlighted light red in Fig 4) with conclusive evidence for a key signal of change (green up arrows for both losses and gains) were from "*cropland*" to "*forest*" and from "*forest*" to "*bare land*" in Tigray and LHZs, from "bare land" to "pasture" in ALL, HSS, WCT, and Tigray, and from "bare land" to "other land uses" in CMC, EDM, EPL, and OTH.

Table 6 shows that forest and bare land (net loss) and pasture (net gain) changed the most. In Tigray, forest (12.8%) lost to bare land (5.01% more) and pasture (6.26%) gained from bare land (2.74% more) than expected randomly. This was similar in ALL, HSS, and OTH, except in OTH where other land use gained from bare land dominantly. Another change was cropland to forest (3.61% more) in Tigray and all LHZs. CMC, EPL, EDM and OTH also lost more bare land to other land uses than expected unlike others.

## Discussion

This study delves into the dynamics of land cover and the determinants of land use allocation by farming households in Tigray, Ethiopia. Utilizing a mixed-methods approach, we combined a plot-level mixed-effects land use allocation model with trend and transition matrix land cover change detection analysis. The ERSS/LSMA-ISA panel data and satellite imagery for 1986 and 2016 served as the foundation for examining land use and land cover changes across Tigray's main livelihood zones (LHZs).

### Land use choice analysis

Our mixed-effects model assumes land use decision is made annually and incorporates a wide range of explanatory variables from different disciplines. The model is estimated using the ERSS/LSMS-ISA panel data, which provides detailed information on land use, land

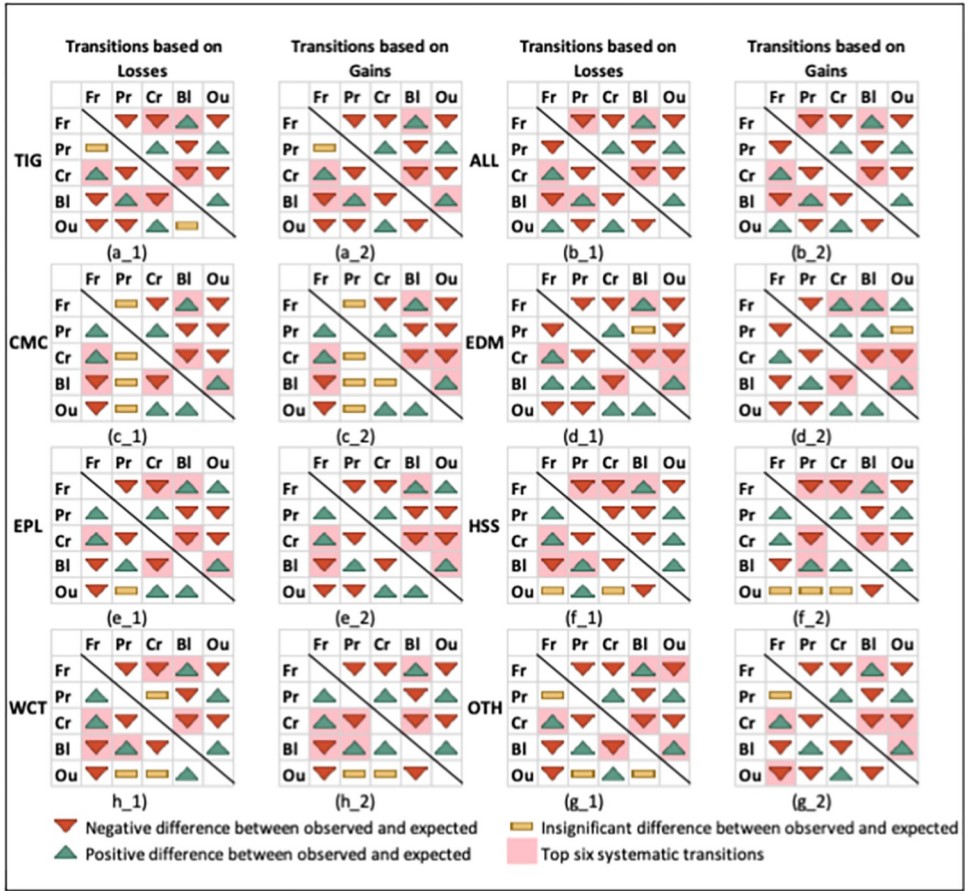

**Fig 4. Transition map of LULC categories in Tigray and its main LHZs from 1986 to 2016.** This shows systematic and random transitions and key signals of change. ALL, Adiyabo Lowland; CMC, Central Mixed Crop; EDM, Enderta Dry Midland; EPL, Eastern Plateau; HSS, Humera Sesame and Sorghum; WCT, West Central Teff; OTH, other land covers; TIG, Tigray. Fr, Forest; Pr, Pasture; Cr, Cropland; Bl, Bare land; and Ou, Other land covers. (a_1) Transitions based on losses for Tigray. (a_2) Transitions based on gains for Tigray. (b_1) to (g_1) Transitions based on losses for each LHZ, ordered as ALL, CMC, EDM, EPL, HSS, WCT and OTH. (b_2) to (g_2) Transitions based on gains for each LHZ, ordered as ALL, CMC, EDM, EPL, HSS, WCT and OTH. Source: Author's own work using ArcGIS and LandSat imagery.

characteristics, household characteristics, and other biophysical features for representative sample of farming households in Tigray, Ethiopia.

A mixed-effects model showed that plot-level factors significantly impacted land use allocation. Plots near residences were often used for forest or other land uses to prevent deforestation by outsiders [74]. Higher-altitude plots were more likely to be fallowed, possibly for animal feed [54]. Soil quality and type influenced land use choices: better soils favored cropland and pasture and vertisols favored fallow land, agreeing with previous studies [75,76]. Tenanted land was more often used for cropland or other land uses than owned land, reflecting intension of tenants to maximize short-term land returns.

Household-level factors also significantly affected land use. More land use types, especially fallow land, were associated with larger and less fragmented plots. This indicates that farmers in Tigray may use most of their newly acquired land for fallowing (1.7 times greater) and leave their more fragmented land idle, both signifying probable capital constraints that impose restrictions on land use [52]. A higher dependency ratio increased fallow land and reduced

**Table 6. Key signals of change by LHZ based on most significant systematic transitions.**

| Transition for the period 1986 to 2016 | Livelihood Zone (LHZ) | | | | | | | |
|---|---|---|---|---|---|---|---|---|
| | **ALL** | **CMC** | **EDM** | **EPL** | **HSS** | **WCT** | **OTH** | **Tigray** |
| *Crop land to Forest* | | | | | | | | |
| Observed (%) | 6.15 | 8.95 | 3.45 | 9.27 | 2.75 | 10.84 | 6.12 | 5.84 |
| Expected (%) | 2.17 (2.35) | 3.83 (4.99) | 1.04 (2.44) | 3.95 (5.52) | 1.12 (2.46) | 4.95 (7.22) | 2.19 (3.66) | 2.23 (3.65) |
| Difference (%) | 3.98 (3.80) | 5.12 (3.96) | 2.42 (1.01) | 5.32 (3.75) | 1.62 (0.29) | 5.89 (3.61) | 3.93 (2.46) | 3.61 (2.19) |
| Difference/Expected | 1.8 (1.62) | 1.34 (0.79) | 2.33 (0.41) | 1.35 (0.68) | 1.45 (0.12) | 1.19 (0.50) | 1.79 (0.67) | 1.62 (0.60) |
| *Forest to Bare land* | | | | | | | | |
| Observed (%) | 18.41 | 9.30 | 4.60 | 9.57 | 20.38 | 15.69 | 9.62 | 12.98 |
| Expected (%) | 10.34 (11.37) | 7.65 (5.68) | 3.73 (2.22) | 6.20 (4.64) | 13.30 (14.52) | 10.79 (9.26) | 5.48 (4.62) | 7.97 (7.21) |
| Difference (%) | 8.07 (7.05) | 1.65 (3.62) | 0.87 (2.37) | 3.37 (4.93) | 7.07 (5.86) | 4.90 (6.43) | 4.14 (4.99) | 5.01 (5.78) |
| Difference/Expected | 0.70 (0.62) | 0.22 (0.64) | 0.23 (1.07) | 0.54 (1.06) | 0.5 (0.40) | 0.40 (0.69) | 0.75 (1.08) | 0.63 (0.80) |
| *Bare land to pasture* | | | | | | | | |
| Observed (%) | 17.07 | | | | 11.64 | 4.48 | | 6.26 |
| Expected (%) | 9.90 (9.77) | | | | 7.69 (7.44) | 1.74 (2.11) | | 3.16 (3.53) |
| Difference (%) | 7.18 (7.31) | | | | 3.95 (4.20) | 2.74 (2.37) | | 3.10 (2.74) |
| Difference/Expected | 0.73 (0.75) | | | | 0.51 (0.56) | 1.57 (1.12) | | 0.98 (0.78) |
| **Bare land to other uses** | | | | | | | | |
| Observed (%) | | 6.66 | 6.29 | 6.04 | | | 6.89 | |
| Expected (%) | | 3.06 (4.25) | 3.57 (3.99) | 2.69 (3.63) | | | 4.58 (3.93) | |
| Difference (%) | | 3.60 (2.42) | 2.72 (2.31) | 3.36 (2.41) | | | 2.31 (2.96) | |
| Difference/Expected | | 1.18 (0.57) | 0.76 (0.58) | 1.25 (0.66) | | | 0.50 (0.75) | |

Note: This table shows the land cover types with top 6 dominant systematic transitions in each LHZ. Percentages are relative to total landscape. Values in brackets are based on gains and values off brackets are based on losses. ALL, Adiyabo Lowland; CMC, Central Mixed Crop; Enderta Dry Midland; EPL, Eastern Plateau; HSS, Humera Sesame and Sorghum; WCT, West Central Teff; OTH, other Livelihood zones.

cropland and other uses, possibly due to labor shortages in agriculture [77]. Higher head education, asset ownership, and climate change perception decreased cropland area, possibly indicating income diversification [57,78,79]. Increased income/wealth increased demand for built-up land similar to other studies [80,81]. Households perceiving climate change may shift their cropland to other categories as climate warming reduces arable land [81].

Wealthier households or those with more livestock tended to use less forest and pasture-land, respectively [82], suggesting that they undervalued the in-situ benefits of forests and preferred larger livestock herds in distant wild grazing areas [83,84]. Market and main road proximity increased the use of pasture, fallow land, and other land uses, indicating that market access and infrastructure availability influenced land use choices by creating opportunities for off-farm activities and livestock production [75].

Product price, annual precipitation, and LHZ also affected land use allocation at the community level. Forest product price increased forest land use, possibly reflecting the profitability of forest products and the conservation incentive [72]. Annual precipitation decreased forest land use, possibly due to the suitability of wetter areas for agriculture in arid Tigray. Land use allocation differed across different LHZs, reflecting biophysical and socio-economic differences among them.

However, though the model obtained high level of explanation for the recent changes in land use, it has a low temporal complexity covering only a short duration (5 years) and a few time steps (3 years). This limits the ability of the model to capture the long-term trends and patterns of land use change [50].

## Land cover change

Assuming land use transition implied land cover change, and vice versa, the study analyzed historic satellite data (1986 and 2016) to examine land cover change trajectories, complementing the mixed-effects model of land use allocation determinants. The years of special data were chosen based on the region's history (the famous 1986 drought), the panel data's end year (2016), and the data availability issues (satellite image validation problems for other years due to size and security). The study uses a similar land cover classification scheme that matches the land use classification of plots in the ERSS/LSMA-ISA dataset. It also acknowledges the limitations of the satellite data in capturing land cover heterogeneity and diversity within pixels, detecting intermediate land cover changes, accounting land cover changes for recent developments that arise from spatial resolution and temporal frequency mismatches, but the satellite data can still capture the dominant land cover type reducing the noise and uncertainty caused by the spatial resolution mismatch [67].

This change detection analysis revealed LULC dynamics in Tigray and its main LHZs from 1986 to 2016, in line with Betru [85]. The results also indicated that the most alarming change was the systematic conversion of forest (12.8%) to bare land in Tigray and all LHZs except for CMC, where forest cover increased. Forest cover decreased by 18.7% (0.7% per annum) in Tigray, resulting in a net loss of 5.7% of the total landscape. Forest degradation was most rapid in HSS (net loss of 18%), followed by ALL (net loss 10%). This indicates that forest was the most vulnerable and dynamic LULC category in Tigray, followed by pasture. These findings agree with previous studies that reported the ongoing deforestation and forest degradation in Tigray [86,87] and Ethiopia [10,88]. This threatens the environment and the livelihoods of local communities through losses in soil quality, water availability, biodiversity, carbon sequestration, food, and resources. Solomon et al. (2019) stressed that forest cover dynamics in Tigray can affect ecosystem services and livelihood losses.

Besides forest loss, another notable change was the expansion of pasture (6.26%) into bare land in Tigray and most LHZs except for CMC and EDM, where pasture was abandoned. Pasture increased by 103.6%, resulting in a net gain of 5.6% of the total landscape. Pasture expansion was most rapid in ALL and HSS (net gain of 16.5% and 15.8%). However, this did not offset the loss of forest cover to bare land. Cropland was relatively stable (almost zero net loss) but with a swap of 14% of the total landscape.

Despite these unfavorable changes, Farming households in Tigray and all LHZs showed commitment to systematic cropland afforestation, but only CMC saw a net increase in forest cover (8%) by afforesting cropland and pasture. Several studies back cropland-to-forest change in Tigray and Amhara [17,21]. EDM also changed bare land to other uses, likely due to urban growth around Mekelle city and other growing towns in the LHZ.

## Policy implications

Based on the findings, this study suggests some policy measures for sustainable land use planning and management in Tigray and its livelihood zones:

- Supports and Incentives: To promote forest benefits, the government should provide market linkages, roads, facilities, services, forestry subsidies, and land tenure security for farming households. These can boost their income, food security, and resilience, and foster forest conservation and restoration [89]. Studies revealed that market linkages can reduce rural poverty and hunger [90], forest subsidies can help farmers access markets, credit, and insurance for forest products and services [91], and land tenure security can enhance land resource investment and management [92,93].

- Balancing livelihoods and conservation: Policy-makers need to balance the environmental and local community needs, while appreciating efforts to regenerate ecosystems in croplands/pastures. Therefore, cropland and pasture reductions observed in favor of forest should be managed carefully as cropland/pasture reduction should be optimal while allowing ecosystems to recover [94,95].

- Land Consolidation Programs: To improve land use efficiency and productivity, government should implement land consolidation programs that reorganize the plots of farmers into larger and more contiguous units, while keeping the same land rights and ownership. Land consolidation programs can also facilitate irrigation, drainage, and other infrastructure, and the adoption of improved land management practices that improve income, food security, and resilience of land users [96,97].

- Participatory Approaches: To prioritize forest regeneration on bare land in reafforestation efforts, the government should involve local communities and stakeholders in the planning, implementation, monitoring and evaluation. Participatory approaches can also improve the capacity, ownership, and accountability of local actors, and the social cohesion [98,99].

- Further Analysis: Land use and land cover changes are complex and dynamic processes that are influenced by multiple factors in multiple dimensions. Therefore, further analysis that requires the use of advanced and comprehensive modeling tools that can capture the all the dimensions is needed.

## Conclusion

Using socioeconomic panel and Landsat data, this study examined how various factors influenced land use choice and detect long term trends and patterns of land cover in Tigray and its main livelihood zones from 1986 to 2016. The study showed that land use choice depended on plot, household, and community characteristics, such as plot distance, size and number, household income, forest price, and rainfall. It was also implied that land use allocation was affected by capital constraints that exist, with more fallowing newly acquired land and leaving more fragmentated land idle. This damaged livelihoods.

The study also detected the most important environmental changes in the study area. One was the fast and systematic forest loss (12.8%) to bare land. Forest loss was worst in HSS and ALL zones. The only exception was CMC zone, where forest increased due to cropland and pasture conversion to forest in form of plantation. This means that some farmers gave up their cropland for afforestation, despite its negative impact on their food and income. Another was the fast and systematic pasture gain (6.26%) from bare land. Pasture gain was best in HSS and ALL zones. However, pasture loss happened in CMC and EDM zones. Moreover, forest to bare land conversion was more than bare land to pasture conversion, endangering both the environment and livelihoods.

## Supporting information

**S1 Fig. Methodology of land use land cover change analysis in Tigray, Ethiopia using plot level panel data and Landsat satellite imagery.**
(TIF)

**S1 Table. Livelihood Zone of Tigray Region, Ethiopia, based on FEWS NET classification.**
(DOCX)

**S2 Table. LULC classification accuracy for 1986 and 2016.**
(DOCX)

**S3 Table. Results of alternative regression models for land use allocation.**
(DOCX)

**S4 Table. Land use land cover transition matrix by main LHZ and Tigray.**
(DOCX)

**S1 Appendix. Theoretical model and notation for smallholder farming household profit maximization.**
(DOCX)

**S2 Appendix. Model specification and estimation procedure for land use allocation.**
(DOCX)

**S3 Appendix. Land use land cover detection techniques.**
(DOCX)

**S4 Appendix. A household survey questionnaire.**
(DOCX)

**S1 Dataset. Minimal data that support the findings of this study.**
(RAR)

## Acknowledgments

I am deeply grateful to my friend and colleague Mantta Yonatan Desalegn (PhD) from Wolaita Sodo University for his academic and personal support throughout my research journey. He has shared his valuable insights and discussions with me. I also thank Obbo Addisu Adamte (PhD) for his substantial insights and discussions in ArcGIS handling. Both of them have encouraged me with their courage and optimism in times of dire challenges and difficulties.

## Author Contributions

**Conceptualization:** Tadele Tafese Habtie.

**Data curation:** Tadele Tafese Habtie.

**Formal analysis:** Tadele Tafese Habtie.

**Methodology:** Tadele Tafese Habtie.

**Software:** Tadele Tafese Habtie.

**Supervision:** Ermias Teferi, Fantu Guta.

**Visualization:** Tadele Tafese Habtie.

**Writing – original draft:** Tadele Tafese Habtie.

**Writing – review & editing:** Ermias Teferi, Fantu Guta.

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
