## [Decision Letter · Decision Letter 0]

22 Nov 2023

PONE-D-23-33020Multi-level determinants of Land Use Land Cover Change in Tigray - A Mixed-Effects Approach Using Socioeconomic Panel and Satellite DataPLOS ONE

Dear Dr. Habtie,

Thank you for submitting your manuscript to PLOS ONE. After careful consideration, we feel that it has merit but does not fully meet PLOS ONE’s publication criteria as it currently stands. Therefore, we invite you to submit a revised version of the manuscript that addresses the points raised during the review process.

We look forward to receiving your revised manuscript.

Kind regards,

Josily Samuel, Ph.D

Academic Editor

PLOS ONE

Journal Requirements:

4. We note that Figures 1, 2a and 2b in your submission contain [map/satellite] images which may be copyrighted. All PLOS content is published under the Creative Commons Attribution License (CC BY 4.0), which means that the manuscript, images, and Supporting Information files will be freely available online, and any third party is permitted to access, download, copy, distribute, and use these materials in any way, even commercially, with proper attribution. For these reasons, we cannot publish previously copyrighted maps or satellite images created using proprietary data, such as Google software (Google Maps, Street View, and Earth). For more information, see our copyright guidelines: http://journals.plos.org/plosone/s/licenses-and-copyright.

     1. You may seek permission from the original copyright holder of Figures 1, 2a and 2b to publish the content specifically under the CC BY 4.0 license.  

Reviewers' comments:

Reviewer's Responses to Questions

**Comments to the Author**

1. Is the manuscript technically sound, and do the data support the conclusions?

Reviewer #1: Yes

Reviewer #2: Partly

2. Has the statistical analysis been performed appropriately and rigorously? 

Reviewer #1: Yes

Reviewer #2: I Don't Know

3. Have the authors made all data underlying the findings in their manuscript fully available?

Reviewer #1: Yes

Reviewer #2: No

4. Is the manuscript presented in an intelligible fashion and written in standard English?

Reviewer #1: Yes

Reviewer #2: No

5. Review Comments to the Author

Reviewer #1: 1. Line 79: I am not as such convinced why religion was described as it is not relevant to this particular topic

2. Line 89-94: is out of the context of this study and hence remove

3. The 17 livelihood zones and enumeration areas should have been briefly described for readers and supported with relevant literature. Please provide some background information with regard to these two important concepts indicated

4. Line 98: change "farms" in to " farmers" which It seems that farms are not engaged.

5. Line 106-109: Do not you think that soil and water conservation initiatives are one of the components of PSNP? Please try to show the difference if it is actually different? For me it is part of the SLM programs initiated through PSNP

6. Line 126-127: when was the survey conducted? Did you get the same households? Please clarify

7. Line 152: How many key informant interviewees were used and why?

8. Line 210-215: repeated

9. The discussion section must be organized in to sub heading similar to the results section

Reviewer #2: Land tenure is an important topic globally and specifically in the context of Africa due to the different types of land tenure and its effects on land use and land use changes. Land tenure is still underresearched. It looks like if the authors did an extensive data collection and data analysis via household and spatial information. Even though this approach is appreciated, a few things are unclear for me: the overall research question/aim of the study, the specific years where data were used for the analysis (e.g. socio-economic data for 1986), specific methods (details are missing), among others.

Introduction: The Introduction is too short. There should be more content related to similar work of LULCC in the Tigray Region. There are already quite a few publications about LULCC in the Tigray Region. You have tried to provide your arguments (what distinguishes your research from others) but this could be better filled with content. I understand that your manuscript is already quite long but it might be better to shift other content to the annex and rather improve the introduction.

Background: The map needs to be improved (see my comments in the manuscript).

Method: There are existing sources of data that were used (ERSS and LSMS-ISA) but there is not much explanation. Please provide more details about the identification of the representative sample of households, collected variables and their definition, e.g. survey template, reason of data collection, time of the data collection, etc. The research question in “Method of data analysis” does not fit to the objective from the introduction. I am confused what is actually the aim of this study.

Only two years of spatial data were compared (1986 and 2016). The spatial classification of land use types is unclear and needs more specification that is backed-up with references. In addition, no data were available for the year 1986 from the Rural Socioeconomic Survey (only 2012, 2014, and 2016). How this data gap was addressed?

Results: Figure 2c, 3 and 4 are difficult to read and hard to understand. This rather causes confusion. Actually, I stopped reading until the end of the results´ section because it is hard to understand.

Discussion: The methodological discussion is missing. The most recent spatial data is from 2016 - it means, it is 7 years ago and might not reflect current land use changes (current realities). I recommend to use updated spatial information.

Conclusion: The section “Policy implication” is out of place. A conclusion should not show new content. Please add this section to the discussion, give more explanation and support it with references.

References are largely missing that the reader cannot judge about the quality of the work. Information is partially missing, too (especially regarding the spatial land use classification approach and the detection of land use changes).

And please address my comments in the manuscript.

6. PLOS authors have the option to publish the peer review history of their article (what does this mean?). If published, this will include your full peer review and any attached files.

Reviewer #1: **Yes: **Terefe Tolessa Muleta

Reviewer #2: No

---

## [Author Response · Author response to Decision Letter 0]

5 Jan 2024

Dear Dr. Josily Samuel,

We are grateful for your and reviewers’ constructive feedback on our manuscript entitled “Multi-level determinants of land use land cover change in Tigray, Ethiopia: A mixed-effects approach using socioeconomic panel and satellite data” (manuscript number PONE-D-23-33020), submitted to PLOS ONE.

We appreciate your consideration of our work for revision. We have revised the manuscript according to your comments and addressed all your concerns.

This letter explains how we have revised our manuscript in response to each of your comments. We hope that our revisions meet your expectations, and our manuscript is now ready for publication in PLOS ONE.

We have prepared a point-by-point response to each issue raised by you and the reviewers. We have also listed the changes that we have made in the revised manuscript, which are marked in blue. For the comments that only appear in the manuscript (mainly related to reference, definition of terms, acronym style, grammar, or formatting), we have summarized them in annexed Table 1 that includes the reviewer comment, the change made, and the location in the revised manuscript and justification (if needed). We do not categorize the comments to reviewer #1 and #2, as there is no clear indicator whose comments are in the manuscript. We apologize for this.

Please find below our detailed responses to you and the reviewers, followed by the list of changes that we have made in the revised manuscript.

Sincerely,

Tadele Tafese Habte

Centre for Environment and Development, College of Development Studies, Addis Ababa University, Addis Ababa, Ethiopia

Response to Reviewer #1

Reviewer comment 1 

• Line 79: I am not as such convinced why religion was described as it is not relevant to this particular topic

Author response 1

Thank you for your suggestion. We agree that religion is not relevant to our study on land use choice and land cover change. We have removed the sentence about the respondents’ religious affiliation from the background section, paragraph 1. 

Reviewer comment 2

• Line 89-94: is out of the context of this study and hence remove

Author(s) response 2

Thank you for your feedback. We acknowledge that the paragraph was not relevant to our study on land use/land cover change, as it did not match the temporal scope or the variables of our analysis. Therefore, we have removed the paragraph from the background section.

Reviewer comment 3

• The 17 livelihood zones and enumeration areas should have been briefly described for readers and supported with relevant literature. Please provide some background information with regard to these two important concepts indicated.

Author(s) response 3

We think this is good suggestion. We have added a short description of the enumeration areas (EA) and the livelihood zones (LHZ) in lines 83-96. The LHZ is a spatial classification based on the dominant livelihood activities and agro-ecological conditions, while the EA is the primary sampling (smallest geographic) unit, used by the Central Statistics Agency (CSA) of Ethiopia for survey sampling and data availability. We have cited the relevant sources for these definitions. 

Reviewer comment 4

• Line 98: change "farms" in to " farmers" which It seems that farms are not engaged.

Author(s) response 4

Thank you for your correction. We have changed the word “farms” to “farmers” in line 99. 

Reviewer comment 5

• Line 106-109: Do not you think that soil and water conservation initiatives are one of the components of PSNP? Please try to show the difference if it is actually different? For me it is part of the SLM programs initiated through PSNP.

Author(s) response 5

Thank you for your comment. We agree that SWC is one of the SLM programs implemented through PSNP. However, we also want to highlight the SWC campaign that the government of Tigray has launched to restore degraded lands, which involves the participation of all farmers regardless of their PSNP status. We have added a sentence to clarify this difference in lines 108-109. 

Reviewer comment 6

• Line 126-127: when was the survey conducted? Did you get the same households? Please clarify

Author(s) response 6

Thank you for your inquiry. We have added more details on the follow-up survey that we conducted to supplement the ERSS, which is part of the LSMS-ISA project by the World Bank. This survey reached out to all the 387 households who participated in the 3rd round of ERSS/ LSMS-ISA in August 2019. We have mentioned this information in lines 137-141.

Reviewer comment 7

• Line 152: How many key informant interviewees were used and why?

Author(s) response 7

To validate the old LULC maps, we interviewed 34 key informants (one from each EA) who were elderly people. The number of interviews was limited by the resource and access constraints, as we conducted them along with the ERSS/LSMS-ISA follow-up survey. This is indicated in lines 175-176. 

Reviewer comment 8

• Line 210-215: repeated

Author(s) response 8

Thank you for your observation. We have removed the duplicate paragraph from page 12, under empirical model and estimation. 

Reviewer comment 9

• The discussion section must be organized in to sub heading similar to the results section

Author(s) response 9

Thank you for your suggestion. We have followed your recommendation and divided the discussion section into three subsections: (1) land use choice analysis, (2) land cover change, and (3) policy implications, matching the results section. Introduced subsection headings are indicated in lines 425, 464 and 500, respectively.

Response to Reviewer #2

Reviewer comment 1

• Introduction: The Introduction is too short. There should be more content related to similar work of LULCC in the Tigray Region. There are already quite a few publications about LULCC in the Tigray Region. You have tried to provide your arguments (what distinguishes your research from others) but this could be better filled with content. I understand that your manuscript is already quite long but it might be better to shift other content to the annex and rather improve the introduction.

Author(s) response 1

Thank you for your constructive feedback. We understand your concern about the introduction section being too short and lacking sufficient content on LULC change in Tigray. We have revised the introduction section by adding more references on LULC change in Tigray and highlighting the research gaps that our study addresses. We have also moved some content to the annex to shorten the manuscript. Please see lines 57-66 for the revised introduction section.

Reviewer comment 2

• Background: The map needs to be improved (see my comments in the manuscript).

Author(s) response 2

Thank you for your feedback on the map of the study area. You wanted us to define the enumeration area (EA) and the land use (LU) types, and to show the LU types on the map. We have updated our manuscript to explain the EA and the livelihood zones (LHZ) on the map. The EA is the smallest geographic unit used by the central statistics agency (CSA) of Ethiopia for survey sampling and data availability. The LHZ is a spatial classification based on the dominant livelihood activities and agro-ecological conditions. We have added these definitions to the caption of Fig. 1 in lines 83-87 and to the introduction section in lines 88-96. LU types in this study are the main land purposes of plots (cropland, forest, pasture, fallow and other uses) as defined in the Ethiopia Rural Socioeconomic Survey/ World Bank’s Living Standards Measurement Study, Integrated Survey on Agriculture (ERSS/LSMS-ISA) dataset. We do not show plot level LU types on the map due to confidentiality terms of using the ERSS/LSMS-ISA. We have defined LU types in lines 229-230. We have removed the map of land cover types, as requested by the editor, due to some copyright issues. We apologize for this inconvenience and thank the editor for noticing it. We hope that this answers your questions and that you are happy with our revisions.

Reviewer comment 3

• Method: There are existing sources of data that were used (ERSS and LSMS-ISA) but there is not much explanation. Please provide more details about the identification of the representative sample of households, collected variables and their definition, e.g., survey template, reason of data collection, time of the data collection, etc. The research question in “Method of data analysis” does not fit to the objective from the introduction. I am confused what is actually the aim of this study.

Author(s) response 3

Thank you for your feedback. We have revised the sub-section on data source and method of data analysis (lines 115-141) to provide more information about the socioeconomic data, both ERSS and its follow-up survey, which captured the public perceptions of climate and LULC change. We have explained the following aspects of the surveys: (1) their aim and scope, (2) their sampling strategy and size, (3) their main variables and definitions (or survey templates in S4_Appendix), (4) their timing and rationale, and (5) their availability and access. We are sorry for the inconsistency between the objective in the method of data analysis and in the introduction. This was a mistake that we made while editing the manuscript. We have fixed it in line 190, to match the objective in the introduction. We hope that this clarifies your concerns and that you are satisfied with our revisions.

Reviewer comment 4

• Only two years of spatial data were compared (1986 and 2016). The spatial classification of land use types is unclear and needs more specification that is backed-up with references. In addition, no data were available for the year 1986 from the Rural Socioeconomic Survey (only 2012, 2014, and 2016). How this data gap was addressed?

Author(s) response 4

Thank you for your feedback. We have revised the manuscript to address the issues you raised about the spatial data and the land use classification. We admit that using only two years of spatial data (1986 and 2016) and having no data for 1986 from the ERSS may affect the validity of our analysis. We justify our choices and solutions as follows: First, we encountered two challenges when analyzing LULC change with the panel data from ERSS/LSMS-ISA for 2012, 2014, and 2016. The panel data enabled us to study land use allocation with a mixed-effects model, which explained the recent changes in land use well. However, the panel data had a low temporal resolution (5 years and 3-time steps), which limited our ability to capture the trends and patterns of land use transitions. Therefore, we used spatial data from satellite imagery (LandSat) for 1986 and 2016 for the whole region, to detect more reliable and noticeable land cover change. This created a temporal and spatial mismatch with the land use allocation analysis. We addressed this by combining the long-term land cover change analysis with the short-term land use allocation analysis, while keeping our objective (to examine the determinants of land use choices and land cover dynamics in Tigray). We acknowledged the limitation of this approach in lines 249-258. Second, we selected 1986 and 2016 as the two years of spatial data for these reasons (lines 147-150). We matched 2016 with the year of the panel data (2012-2016). We chose 1986 because it was a drought year in Tigray, Ethiopia, which contrasted with the current situation. Another reason was the difficulty of satellite image processing for such a large area. We wanted to add more years in between, but the war in Tigray in 2020 and the communication blackout stopped us from doing so. We decided to continue with our analysis with the data we had for about 3 years of crisis. Third, we explained the spatial classification of land use types in depth and added more references to support it in lines 143-188. We had land use classification from the panel data at the plot level as cropland, forest, pasture, fallow, and other uses, but the classification from satellite imagery processing was: cropland, grassland/pasture, forest, bushland, bare land, and others. To harmonize the classifications, we used information from the key informant interviews (KII) during the ERSS/LSMS-ISA follow-up survey. The informants said that Natural and plantation forests found in the study area include woodlands, shrubs and bushlands that are under government and communal protection and agroforestry areas. We agreed with the respondents’ definition of forest and included bushland and shrubs in it. 

Reviewer comment 5

• Results: Figure 2c, 3 and 4 are difficult to read and hard to understand. This rather causes confusion. Actually, I stopped reading until the end of the results´ section because it is hard to understand.

Author(s) response 5

Thank you for your feedback. We are sorry for the confusion caused by the graphs of land cover change analysis in the results section. We have made the following changes to improve our presentation: (i) We focused on the change detection based on trend and transition matrix, as suggested by the reviewer, instead of change detection based on change in land cover share. We also deleted the land cover maps, as requested by the editor, due to some copyright issues. (ii) We separated the trend change in land cover (previously Fig. 3a) from the net change (previously Fig. 3b) and made a standalone figure (Fig. 2). We combine net change in land cover (Previously Fig. 3b) and swamp and total change in land cover (Previously Fig. 4a) and made them a standalone figure (Fig. 3). Fig. 2 and Fig. 3 show consistent patterns of land cover change, while Fig. 3 also shows additional information components of change (net, swap, and total changes). (iii) We replaced the persistence indices of each land cover type (previously Fig. 4b) with a table (Table 5) to make it easier to compare the values. You can see the changes made in (i) and (ii) in the new Fig. 2 and Fig. 3 and their captions in lines 342-346 and lines 357-362, respectively. The change made in (iii) is indicated in lines 378-380. (iv) We also rounded the values reported on figures to one decimal place. This may cause some minor changes in numbers. We also rewrite sentences in the results section to match the new presentation format (lines 340-341 and line 350-356). We hope that these changes will make our results clearer and easier to understand.

Reviewer comment 6

• The methodological discussion is missing. The most recent spatial data is from 2016 - it means, it is 7 years ago and might not reflect current land use changes (current realities). I recommend to use updated spatial information.

Author(s) response 6

Thank you for your constructive feedback. We have updated our manuscript to include more methodological discussion in the discussion section. We have added four new paragraphs that cover the following points: (i) The model and the method we used to analyze land use allocation and land cover change, and the data we used for this purpose (lines 418-424). (ii) A short description of the mixed effects model (its assumptions) and the ERSS/LSMA-ISA data we used to fit the model (lines 426-430), and the model’s limitation in capturing long term land use change trends and patterns (lines 460-463). (iii) A short explanation of the approach we followed to detect land cover change, the data we used, and the challenges and limitations of using satellite data, such as spatial resolution and temporal frequency mismatches, limited ability to capture land cover heterogeneity and diversity within pixels, and intermediate land cover changes (lines 465-476). (iv) We added relevant literature citations to support each paragraph. We understand your concern about the timeliness of the spatial data. We selected the 2016 data because it matched the last year of the ERSS/LSMA-ISA panel data, which was the main source of information for the land use allocation determinants. However, we recognize that the ERSS/LSMA-ISA for 2018/19 is a new panel and is not a continuation of the previous ESS waves (Ethiopia - Socioeconomic Survey 2018-2019 (worldbank.org)). Moreover, we had difficulties in getting and validating satellite imagery data for other years due to the large area coverage and security issues in th

---

## [Decision Letter · Decision Letter 1]

2 Feb 2024

PONE-D-23-33020R1

Multi-level determinants of land use land cover change in Tigray, Ethiopia: A mixed-effects approach using socioeconomic panel and satellite data

PLOS ONE

Dear Dr. Habtie,

Thank you for submitting your manuscript to PLOS ONE. After careful consideration, we feel that it has merit but does not fully meet PLOS ONE’s publication criteria as it currently stands. Therefore, we invite you to submit a revised version of the manuscript that addresses the points raised during the review process.

We look forward to receiving your revised manuscript.

Kind regards,

Josily Samuel, Ph.D

Academic Editor

PLOS ONE

Journal Requirements:

Reviewers' comments:

Reviewer's Responses to Questions

**Comments to the Author**

1. If the authors have adequately addressed your comments raised in a previous round of review and you feel that this manuscript is now acceptable for publication, you may indicate that here to bypass the “Comments to the Author” section, enter your conflict of interest statement in the “Confidential to Editor” section, and submit your "Accept" recommendation.

Reviewer #3: (No Response)

2. Is the manuscript technically sound, and do the data support the conclusions?

Reviewer #3: Partly

3. Has the statistical analysis been performed appropriately and rigorously?

Reviewer #3: Yes

4. Have the authors made all data underlying the findings in their manuscript fully available?

Reviewer #3: Yes

5. Is the manuscript presented in an intelligible fashion and written in standard English?

Reviewer #3: No

6. Review Comments to the Author

Reviewer #3: Line no. 79-80 -Agro-climatic zone is a land unit in terms of major climates, suitable for a certain range of crops and cultivars but the name suggests it is based on elevation. Please clarify.

Background - should also include the gap in research which this paper is addressing with the objectives.

Methodology- can you show the whole methodology in flowchart form. Its will make the reader to understand the whole thing better.

Line no. 170- Change the word to Classification from calcification.

Line no. 171- Please tell the year in which you have interviewed the 34-key informant.

Line no. 178- A total of 375 sample points .. its not sampling points its training points for supervised classification to create signature file.

Results

Line no. 281- crop farming can be simply written as farming.

As far as I understood, for LULC you have taken data of 1986 and 2016, while for socio economic study the data is of 2012 to 2016.

its not vertosol, I think its soil order vertisols. Please rewrite the sentence.

Line no. 298- STATA is same the software name you mentioned earlier? If so maintain the same format.

Line no. 401- if you have used 1986 data and 2016 do not write 1986-2016, instead write 1986 and 2016.

7. PLOS authors have the option to publish the peer review history of their article (what does this mean?). If published, this will include your full peer review and any attached files.

Reviewer #3: **Yes: **Pushpanjali

---

## [Author Response · Author response to Decision Letter 1]

11 Feb 2024

Response to Reviewer #3

Reviewer comment 1 

• Line no. 79-80 Agro-climatic zone is a land unit in terms of major climates, suitable for a certain range of crops and cultivars but the name suggests it is based on elevation. Please clarify.

Author response 1

Thank you for your question. We agree that the agroclimatic zonation we used is specific to Ethiopia and may not be applicable to other regions. The agroclimatic background of the region is based on the traditional zonation that reflects the major physical conditions and agricultural land uses according to altitude and crop patterns. Elevation is a key factor that influences temperature and rainfall in Ethiopia. This traditional zonation has been validated by several studies, such as Hurni (1998), who defined the following zones:

o Bereha: hot lowlands below 500 meters, with limited crop production in the arid east and root crops and maize in the humid west.

o Kolla: lowlands between 500 and 1,500 meters, with sorghum, finger millet, sesame, cowpeas, and groundnuts as predominant crops.

o Woina Dega: midlands between 1,500 and 2,300 meters, with wheat, teff, barley, maize, sorghum, chickpeas and haricot beans as predominant crops.

o Dega: highlands between 2,300 and 3,200 meters, with barley, wheat, highland oilseeds, and highland pulses as predominant crops.

o Wurch: highlands between 3,200 and 3,700 meters, with barley as a common crop.

o Kur: highland areas above 3,700 meters, mainly used for grazing pasture.

The dominant agroclimatic zones in Tigray are: Kolla (lowlands), Woina Dega (midlands) and Dega (highlands). 

Reviewer comment 2

• Background - should also include the gap in research which this paper is addressing with the objectives.

Author(s) response 2

Thank you for your valuable feedback. We appreciate your attention to detail. In response to your suggestion, we have now incorporated a paragraph in the background section (lines 111 to 117) that explicitly outlines the research gap this paper aims to address. This addition will provide readers with a clearer understanding of the context and objectives of our study.

Reviewer comment 3

• Methodology- can you show the whole methodology in flowchart form. It will make the reader to understand the whole thing better.

Author(s) response 3

Thank you for your feedback. We have addressed your suggestion by including a flowchart that visually represents the methodology. The flowchart has been added in lines 205 to 206. Additionally, the caption for this flowchart, which is included as supporting information, can be found in the supporting information list at lines 789 to 790. This should enhance the reader’s understanding of the entire process. 

Reviewer comment 4

• Line no. 170- Change the word to Classification from calcification.

Author(s) response 4

Thank you for your observation. We have addressed your comment by replacing “Calcification” with “Classification” in line 176. 

Reviewer comment 5

• Line no. 171- Please tell the year in which you have interviewed the 34-key informant.

Author(s) response 5

Thank you for your feedback. We appreciate your attention to detail. The key informant interviews with elderly individuals were conducted in August 2019, during the ERSS/LSMS-ISA follow-up survey. As a result, we have now included the year in line 182 to provide clarity. 

Reviewer comment 6

• Line no. 178- A total of 375 sample points. It is not sampling points it is training points for supervised classification to create signature file.

Author(s) response 6

Thank you for your comment. The term “sampling points” has been revised to “training points” in line 184 to accurately reflect the purpose of these.

Reviewer comment 7

• Line no. 281- crop farming can be simply written as farming.

Author(s) response 7

Thank you for your valuable feedback. Based on your suggestion, we have replaced “crop farming” with the more concise term “farming” in line 287. 

Reviewer comment 8

• As far as I understood, for LULC you have taken data of 1986 and 2016, while for socio economic study the data is of 2012 to 2016.

Author(s) response 8

Thank you for your valid concern. We have used land cover data for two time points: 1986 and 2016. We have corrected all the instances where we mistakenly wrote “1986 to 2016” instead of “1986 and 2016” when describing the data we used. Please see lines 428 and 472 for the changes. 

Reviewer comment 9

• It is not vertosol, I think its soil order vertisols. Please rewrite the sentence.

Author(s) response 9

We appreciate your observation. You are correct, the soil order is vertisols, not vertosol. We have revised the sentence accordingly in Table 2 and 4 and line numbers 289, 312, and 441. 

Reviewer comment 10

• Line no. 298- STATA is same the software name you mentioned earlier? If so maintain the same format.

Author(s) response 10

We thank you for your attention to detail. You are right, the software name is Stata, not STATA. We have corrected this typo in line 304 and ensured consistent formatting throughout the manuscript. 

Reviewer comment 11

• Line no. 401- if you have used 1986 data and 2016 do not write 1986-2016, instead write 1986 and 2016.

Author(s) response 11

We are grateful for your observation. This is the same issue as in response 8. We have changed “1986 to 2016” to “1986 and 2016” in line 428 and elsewhere as appropriate.

Response to Editor

Editor comment 1

Author(s) response 1

We appreciate your comment and we have carefully reviewed our reference list. All the citations and references are complete and correct. We have used Zotero reference manager to manage our references. Zotero is linked to the Retraction Watch Database and automatically checks for retracted articles with DOI. For the references without DOI, we manually searched the Retraction Watch Database using Author(s) and Title as filtering criteria. We did not find any retracted references in our list. Therefore, we did not need to cite any retracted article or provide any rationale for doing so.

---

## [Decision Letter · Decision Letter 2]

15 Mar 2024

PONE-D-23-33020R2

Multi-level determinants of land use land cover change in Tigray, Ethiopia: A mixed-effects approach using socioeconomic panel and satellite data

PLOS ONE

Dear Dr. Habtie,

Thank you for submitting your manuscript to PLOS ONE. After careful consideration, we feel that it has merit but does not fully meet PLOS ONE’s publication criteria as it currently stands. Therefore, we invite you to submit a revised version of the manuscript that addresses the points raised during the review process.

We look forward to receiving your revised manuscript.

Kind regards,

Josily Samuel, Ph.D

Academic Editor

PLOS ONE

Journal Requirements:

Additional Editor Comments:

Dear authors

As per the reviewers comments, the response from authors are unsatisfactory. Please do take every corrections mentioned by reviewers into account and edit the manuscript thoroughly.

Regards,

Josily

Reviewers' comments:

Reviewer's Responses to Questions

Comments to the Author

1. If the authors have adequately addressed your comments raised in a previous round of review and you feel that this manuscript is now acceptable for publication, you may indicate that here to bypass the “Comments to the Author” section, enter your conflict of interest statement in the “Confidential to Editor” section, and submit your "Accept" recommendation.

Reviewer #3: (No Response)

2. Is the manuscript technically sound, and do the data support the conclusions?

Reviewer #3: Partly

3. Has the statistical analysis been performed appropriately and rigorously?

Reviewer #3: (No Response)

4. Have the authors made all data underlying the findings in their manuscript fully available?

Reviewer #3: (No Response)

5. Is the manuscript presented in an intelligible fashion and written in standard English?

Reviewer #3: (No Response)

6. Review Comments to the Author

Reviewer #3: (No Response)

7. PLOS authors have the option to publish the peer review history of their article (what does this mean?). If published, this will include your full peer review and any attached files.

Do you want your identity to be public for this peer review? For information about this choice, including consent withdrawal, please see our Privacy Policy.

Reviewer #3: No

---

## [Author Response · Author response to Decision Letter 2]

24 Mar 2024

Dear Dr. Josily Samuel,

We are grateful for your and reviewer constructive feedback on our manuscript entitled “Multi-level determinants of land use land cover change in Tigray, Ethiopia: A mixed-effects approach using socioeconomic panel and satellite data” (manuscript number PONE-D-23-33020), submitted to PLOS ONE. We appreciate your consideration of our work for revision.

This letter explains how we have revised our supplementary information named S1_flowchart in response to the reviewer’s comments. We hope that our revisions meet your expectations, and our manuscript is now ready for publication in PLOS ONE.

We have prepared a point-by-point response to each issue raised by you and the reviewer. We have also listed the changes that we have made in the revised supplementary information.

Please find below our detailed responses to you and the reviewer, followed by the list of changes that we have made.

We thank you and the reviewer for your valuable feedback and your time and effort in reviewing our manuscript. We hope that you will find our revisions satisfactory and accept our manuscript for publication in PLOS ONE.

Sincerely,

Tadele Tafese Habte

Centre for Environment and Development, College of Development Studies, Addis Ababa University, Addis Ababa, Ethiopia

Response to Reviewer #3

Reviewer comment 1 

• The reviewer put comments [hand written] on the S1_flowchart attached to email and requested to revise accordingly. 

Author response 1

Thank you for your comments on the flowchart summarizing the methodology. In response to the reviewer’s clear and direct feedback, the S1_flowchart has been revised to incorporate the suggested additions and deletions. The concepts indicated by the reviewer have been integrated, and unnecessary elements have been removed as per the guidance provided. Specifically; 

o We have clarified the types of data utilized: socioeconomic and satellite data, and have removed questioning data availability. Additionally, we have detailed the two sources of socioeconomic data used, namely primary and secondary data sources.

o In response to the comment on land use classification, we acknowledge the hybrid approach taken in our methodology. Initially, we employed unsupervised techniques for class identification, using Iterative Self-Organizing Data Analysis clustering to group pixels based on spectral similarities. Subsequently, ground control points (GCPs) collected via high-resolution imagery from Google Earth and GPS coordinates served as training data for a supervised Maximum Likelihood Classification (MLC) algorithm. This refined the land use categories with established spectral signatures. Although our classification process involved these stages, the final supervised MLC algorithm determined our classification. Therefore, we have specified the application of supervised image classification in the revised flowchart.

o We appreciate your concern regarding the temporal data representation. We have corrected the temporal data representation from “1986 to 2016” to the specific time points: 1986 and 2016, as used in our manuscript.

o We concur that the back loop to the research question is redundant once the research question has been addressed. The flowchart has been updated to remove this loop, aligning with the recommendation. 

We are committed to ensuring the highest quality of our work and are open to further revisions should there be additional feedback.

Response to Editor

Editor comment 1

Author(s) response 1

We thank you for emphasizing the importance of an accurate and reliable reference list. Following your advice, we have conducted a careful review of our references using Zotero, integrated with the Retraction Watch Database, to automatically check for retracted articles with DOIs. Additionally, we performed manual searches for references without DOIs. We can confirm that our reference list is complete, accurate, and free of retracted articles. We are prepared to make any further adjustments if necessary.

---

## [Editor Report · Decision Letter 3]

21 May 2024

Multi-level determinants of land use land cover change in Tigray, Ethiopia: A mixed-effects approach using socioeconomic panel and satellite data

PONE-D-23-33020R3

Dear Dr. Habtie,

We’re pleased to inform you that your manuscript has been judged scientifically suitable for publication and will be formally accepted for publication once it meets all outstanding technical requirements.

Kind regards,

Mohammed Sarfaraz Gani Adnan, PhD

Academic Editor

PLOS ONE
---

## [Editor Report · Acceptance letter]

23 May 2024

PONE-D-23-33020R3 

PLOS ONE

Dear Dr. Habtie, 

I'm pleased to inform you that your manuscript has been deemed suitable for publication in PLOS ONE. Congratulations! Your manuscript is now being handed over to our production team.

Kind regards, 

on behalf of

Dr. Mohammed Sarfaraz Gani Adnan 

Academic Editor

PLOS ONE